# Factors Associated with Farrowing Duration in Hyperprolific Sows in a Free Farrowing System under Tropical Conditions

**DOI:** 10.3390/ani12212943

**Published:** 2022-10-26

**Authors:** Yosua Kristian Adi, Rafa Boonprakob, Roy N. Kirkwood, Padet Tummaruk

**Affiliations:** 1Centre of Excellence in Swine Reproduction, Department of Obstetrics, Gynaecology and Reproduction, Faculty of Veterinary Science, Chulalongkorn University, Bangkok 10330, Thailand; 2Department of Reproduction and Obstetrics, Faculty of Veterinary Medicine, Universitas Gadjah Mada, Yogyakarta 55281, Indonesia; 3School of Animal and Veterinary Sciences, University of Adelaide, Roseworthy, SA 5371, Australia

**Keywords:** climate, litter size, parturition, pig, stillbirth

## Abstract

**Simple Summary:**

Sows in most breeding herds worldwide have larger litters than several years ago. One of the most important problems when using these genetics is the prolonged duration of farrowing, which can cause postpartum complications in sows and increase the percentage of stillborn piglets per litter. In this retrospective study, we found that the farrowing duration of sows kept in a free farrowing system in a tropical environment was associated with several factors. A high number of piglets born per litter, a high parity number, parturition during working hours, and high temperature and humidity in the 7 days before parturition led to a prolonged farrowing duration. In these sows, farrowing was longer than the acceptable farrowing duration, which may cause a higher number of stillborn piglets. Therefore, management for sows during the perinatal period needs to be considered, especially in hyperprolific and older sows, as well as those that farrow during working hours.

**Abstract:**

The ongoing selection for increased litter size has had significant impacts on sow husbandry practice. The present study investigated factors associated with farrowing duration and the proportion of sows that had prolonged farrowing in modern hyperprolific sows kept in a free farrowing system in a tropical environment. Farrowing data from 2493 Landrace x Yorkshire cross-bred sows in a commercial swine herd in Thailand were included in the study. The time of farrowing, parity number, litter size, and the birth status of each piglet were recorded. Farrowing duration was analysed using multiple analyses of variance. Total number of piglets born per litter (TB), parity, and time onset of farrowing were included in the statistical models. On average, TB, piglets born alive, and farrowing duration were 13.7, 12.1, and 221.0 min, respectively. Of these sows, 26.4% had TB ≥ 16 and 21.7% had a prolonged farrowing duration (≥300 min). Farrowing duration was positively correlated with TB (r = 0.141, *p* < 0.001), percentage of stillborn (SB) piglets per litter (r = 0.259, *p* < 0.001), percentage of mummified foetuses (MF) per litter (r = 0.049, *p* = 0.015), piglet birth weight (r = 0.068, *p* < 0.001), and litter birth weight (r = 0.041, *p* = 0.043). The proportion of SB per litter was higher and piglet birth weight lower in litters that had ≥16 TB than those with 8–12 TB (*p* < 0.05). The farrowing duration of sows with parity numbers 5–7 (247.7 ± 5.1 min) and 8–10 (237.1 ± 5.1 min) was longer than that of sows with parity numbers 1 (188.3 ± 5.2 min) and 2–4 (214.3 ± 3.9 min) (*p* < 0.05). Sows that had started farrowing during working hours had longer farrowing durations (229.3 ± 3.6 min) than those that had started farrowing during non-working hours (217.6 ± 3.4 min, *p* = 0.017). In multiparous sows, the duration of farrowing was positively correlated with the maximum temperature (r = 0.056, *p* = 0.012) and the maximum temperature–humidity index (r = 0.059, *p* = 0.008) in the 7 days before farrowing. The present data confirm that TB, sow parity, and time of onset of farrowing are significant risk factors for a prolonged farrowing.

## 1. Introduction

Over the last decade, genetic progress on the litter size of sows had a high impact on sow husbandry practice. As a result, sows in most breeding herds worldwide have larger litters than several years ago [1]. In general, sows with a total number of piglets born per litter (TB) of ≥16 can be regarded as hyperprolific sows [2]. In practice, many sows can reach such a high litter size in many parity numbers [3]. Hyperprolific sow genetics have been widely distributed to the swine industry worldwide, including Thailand [1,2,3,4]. However, one of the most important problems when using these genetics is the prolonged duration of farrowing [5]. Oliviero et al. [6] reported that the average duration of farrowing in modern swine production can range from 156 to 262 min, and farrowing lasting more than 300 min should be considered prolonged farrowing. Likewise, Ju et al. [7] suggested that a farrowing duration of 240 to 300 min can be considered the ideal cutoff point for Landrace x Yorkshire hybrid sows in a commercial pig farm having 5 to 22 TB. Prolonged farrowing duration can cause postpartum complications in sows and increase the percentage of stillborn piglets per litter (SB) [8]. For instance, the SB in sows with a long duration of farrowing (>4 h) is higher than that in sows with a short duration (<2 h) of farrowing (29.2% versus 7.9%, respectively) [8]. The average farrowing duration of sows without any SB is shorter than that of sows with SB ≥ 3 (221.0 versus 373.0 min, respectively) [4]. Therefore, understanding the factors significantly associated with the duration of farrowing in sows under field conditions is important.

Farrowing duration is one aspect that is affected by an increase in litter size in modern sows. The farrowing duration in most of the European domestic pig breeds has increased from about 2 h per 12 TB to 6 h 40 min per 19 TB [5]. However, comprehensive information about the factors related to the duration of farrowing in modern hyperprolific sows in Asia is limited [4]. Factors known to influence farrowing duration are the state of constipation, the body condition of the sow, housing, gestation length, sow age, and genetics [6]. In a tropical environment, hyperprolific sows with an average TB of 17.5 have an average farrowing duration of 330.6 min [4]. Thus, prolonged farrowing duration in hyperprolific sows has become one of the most important issues.

Prolonged farrowing duration can have adverse effects on both sows and piglets [8]. The incidence of intrapartum hypoxia in newborn piglets increases following a prolonged duration of farrowing [5]. Parturition is associated with many physiological processes, including both hormonal and behavioural changes, and is the most painful experience for females [9]. Moreover, the expulsion interval of each piglet is positively associated with SB [4]. As the concentration of colostrum IgG decreases by 50% within 6 h after the birth of the first piglet [10], the access of piglets to good-quality colostrum is reduced in sows with a long farrowing duration. Prolonged parturition may reduce piglet vitality at birth [1,5]. A previous study demonstrated that a higher proportion of piglets that attempted to stand after 5 min (38.5%) died compared to piglets that attempted to stand within 1 min (6.3%) after birth [11]. Moreover, retained placentae and uterine inflammation increase in sows with a long duration of farrowing [12]. Thus, sows with a long farrowing duration may have compromised reproductive performance. Oliviero et al. [13] found that 13% of sows that failed to get pregnant at the first insemination after weaning had a relatively long farrowing duration in their previous farrowing.

In tropical environments, the average environmental temperature generally varies from 20 to 35 °C [14]. On average, the temperature at night is a few degrees lower than that during the day. However, it is still a few degrees above the comfort-zone temperature for a pregnant sow [15]. Although farrowing during the night is favourable because of the lower environmental temperature, this is a nonworking period in most pig farms. During late gestation, sows prefer a temperature of 12.6 to 15.6 °C [15]. Lucy and Safranski [16] found that exposure to heat stress in late gestation in sows resulted in some negative effects, such as reduced piglet birth weight and an increased number of stillborn piglets. Not only high temperature but also high relative humidity and/or a high temperature–humidity index during gestation can reduce TB [17]. Heat stress that comes from the environment not only affects piglet characteristics but also compromises the reproductive performance of sows, e.g., prolonged weaning-to-first-service interval and reduced farrowing rate [18]. Interestingly, the threshold of temperatures leading to a prolonged weaning-to-first-service interval is lower for primiparous than for multiparous sows (17 °C vs. 25 °C), and the threshold temperatures leading to reductions in farrowing rates for gilts, primiparous sows and multiparous sows are 20, 21 and 24 to 25 °C, respectively [18]. However, to our knowledge, the influences of temperature and humidity in the tropical environment on farrowing duration in both sows with normal litter size and hyperprolific sows have never been reported. Therefore, we investigated the factors influencing farrowing duration and the proportion of sows that had a prolonged farrowing duration in modern hyperprolific sows in a free farrowing system under tropical conditions. The influences of the time of the onset of farrowing (working vs. non-working hours) and temperature and relative humidity inside the farrowing house for 7 days before farrowing on farrowing duration and the incidence of sows with prolonged farrowing duration were also investigated.

## 2. Materials and Methods

### 2.1. Study Design

The present study included farrowing data from 2493 Landrace x Yorkshire cross-bred sows that farrowed during the period from January to April 2021 in a commercial swine herd in the central region of Thailand. Sows were randomly distributed according to TB into three groups: 8–12 (*n* = 853), 13–15 (*n* = 983) and ≥16 piglets/litter (*n* = 657). Parity number of sows was classified into four groups: 1 (*n* = 506), 2–4 (*n* = 919), 5–7 (*n* = 534) and 8–10 (*n* = 534). The time when the onset of farrowing occurred was classified into two groups: working hours (0700 h–1700 h) (*n* = 1176) and non-working hours (1701 h–0659 h) (*n* = 1317). The average temperature, humidity and temperature–humidity index (THI) during the 7-day period before farrowing were recorded for each individual sow. Farrowing duration was expressed as either a continuous trait (the interval from the first piglet to the last piglet delivered in minutes) or a categorical trait (the proportion of sows that had a farrowing duration of longer than 300 min). Factors including TB, parity number, the time when the onset of farrowing occurred and the average temperature, humidity and THI during the 7-day period before farrowing were analysed to determine their association with the farrowing duration and the proportion of sows that had a prolonged farrowing.

### 2.2. Data: Inclusion and Exclusion Criteria

The original farrowing data were obtained from 2750 sows. The data included sow identity, breed of sows, parity number, date of farrowing, TB, BA, SB, percentage of mummified foetuses per litter (MF), liveborn piglet birth weight, the time when the onset of farrowing occurred and the end of parturition. The farrowing duration for each individual sow was calculated, defined as the period from the first to the last piglet delivery in minutes. Data were scrutinised for correctness, and data with values too extreme were excluded from the analyses. Errors in the reported farrowing times records were checked by calculating the farrowing duration and constructing the frequency distribution of the farrowing duration. The data of sows with too short farrowing duration (<30 min, *n* = 15) and farrowing duration too long (>720 min, *n* = 30) were excluded from the analyses. Old sows (parity numbers ≥ 11, *n* = 7) and sows that had a TB ≤ 7 (*n* = 205) were excluded. In total, 9.4% (*n* = 257) of the raw data were excluded. Thus, the analysed data contained observations on 2493 sows.

### 2.3. Housing and General Management

The sows and gilts were kept in a group housing system during gestation. The number of sows per group was 40, and the size of the gestating pen was 9 × 11 m. The sows were kept group-housed within 3 days after the last insemination until 108 days of gestation before transfer to the farrowing pen. The average daily minimum to maximum temperatures inside the barn during the experimental period were 26.5 ± 0.9 °C (range 21.8–28.2 °C), and the average daily minimum to maximum humidity levels inside the barn were 71.0 ± 1.0% (range 69.0–73.7%). During the first, middle, and late periods of gestation, the sows were fed approximately 1.8–2.0, 2.0–2.2 and 3.0–3.5 kg feed per sow per day, respectively. Three days before the estimated day of farrowing, the feed was reduced to 2.5–3.0 kg of feed per sow per day. The gestation diet contained 15.0% crude protein, 2700 kcal/kg metabolizable energy, and 0.7% lysine. After farrowing, the sows were fed ad libitum. The lactating sows were fed using an automatic feeding machine that allowed the sows to consume feed freely. The lactation diet contained 16.0% crude protein, 3600 kcal/kg metabolizable energy, and 0.8% lysine. Water was provided ad libitum via a drinking nipple. At 109 ± 2.0 days of gestation, sows and gilts were moved to a free farrowing pen system. Temperature and humidity in the farrowing house were recorded manually by stock persons three times a day at 0600 h, 1300 h and 1600 h. The farrowing process was carefully monitored by stock persons in the barn for 24 h daily. The time onset and the end of farrowing, birth weight of live born piglet and the status of the piglets at birth (i.e., live-born, stillborn, or mummified foetuses) were recoded. During farrowing, the sows and gilts were disturbed as little as possible. Single-dose administration of 20 IU oxytocin via intramuscular route (Phenix Pharmaceuticals N.V. Co., Ltd., Hoogstraten, Belgium) was performed if the sow had a birth interval of >60 min and/or no sign of uterine contraction. In addition, 20 IU oxytocin was routinely administered via intramuscular route to all sows after the 10th piglet was born to initiate placental expulsion and milk let-down. Live-born piglets were weighed individually. Stillborn piglets and mummified foetuses were removed and distinguished based on their skin colour, sucked eyes and skin appearance. Stillborns were dead piglets that had pink skin, non-sucked eyes and wet skin, whereas mummified foetuses were dead piglets with dark skin colour, sucked eyes and dry skin. At the end of parturition, all sows were treated with an antipyretic drug (ketoprofen 3.0 mg/kg intramuscularly using Ketaprofen^®^, KELA N.V., Hoogstraten, Belgium). The health status of the sows was monitored routinely by the herd veterinarian. Gestating sows were vaccinated against foot and mouth disease (AFTOPOR^®^, Merial SAS, Lyon, France) and porcine epidemic diarrhoea virus (SUIT-SHOT PT-100^®^, Choong Ang Vaccine Laboratories Co., Ltd., Deajeon, Korea), porcine circovirus (Circumvent PCV^®^, Merck Animal Health, Kenilworth, QL, USA) and Aujeszky’s disease virus (Porcilis^®^ Ad Begonia, Merck Animal Health, Madison, WI, USA). After farrowing, the sows were vaccinated against classical swine fever (Ceva-Phylaxia Veterinary Biologicals Co., Ltd., Budapest, Hungary) and porcine parvovirus–*Leptospira*–erysipelas (Zoetis ZA, Sandton, South Africa). The piglets were vaccinated against *Mycoplasma hyopneumoniae* (Hyogen^®^, Ceva Santé Animale S.A, Libourne, France) at 18–22 days of age.

### 2.4. Statistical Analysis

All statistical analyses were carried out using SAS version 9.4 (SAS Inst., Cary, NC, USA). Descriptive statistics for continuous and categorical variables were analysed using MEANS and FREQ procedures, respectively. Frequency distribution for the duration of farrowing was analysed using the FREQ procedure of SAS. Pearson’s correlation was calculated to determine the associations between farrowing duration and other continuous traits including TB, BA, SB, MF, piglet birth weight, litter birth weight, percentage of piglets with birth weight <1.0 kg, temperature, humidity and THI during the period of 7 days before parturition. Farrowing duration was analysed using multiple analyses of variance, applying the general linear model procedure of SAS. The factors included in the statistical models were TB classes (8–12, 13–15 and ≥16), parity number classes (1, 2–4, 5–7 and 8–10), time of the onset of farrowing (working hours and non-working hours) and two-way interactions with *p <* 0.10. Least-square means were obtained from each variable class and compared using the Tukey–Kramer adjustment for multiple comparisons. The proportion of sows with prolonged farrowing (i.e., >300 min) was expressed as a percentage and analysed using logistic regression in the generalised linear mixed model (GLIMMIX) procedure of SAS. The factors included in the statistical models were TB classes (8–12, 13–15 and ≥16), parity number classes (1, 2–4, 5–7 and 8–10), onset of farrowing (working hours and non-working hours) and two-way interactions with *p* < 0.10. Least-square means were obtained from the models and compared using the Tukey–Kramer adjustment for multiple comparisons. Additionally, the influences of temperature, humidity and THI during 7 days before farrowing on farrowing duration was analysed using the general linear model procedure of SAS. The factors included in the statistical models were TB classes (8–12, 13–15 and ≥16), parity number classes (1, 2–4, 5–7 and 8–10) and two-way interactions with *p* < 0.10. The temperature, humidity and THI values during the 7-day period before parturition were included in the statistical models one at a time, as they were highly correlated. For all analyses, differences at *p* < 0.05 were regarded statistically significant.

## 3. Results

### 3.1. Descriptive Data

Table 1 shows the descriptive statistics on sow reproductive performances and farrowing characteristics. On average, sows in the present study had 13.7 TB with farrowing duration of 30.0 to 716.0 min (Table 1). The proportions of sows that had 8–12, 13–15 and ≥16 TB were 34.2%, 39.4% and 26.4%, respectively. The proportion of sows with the onset of farrowing during working hours and non-working hours were 47.2% and 52.8%, respectively. Of all sows, 21.7% had a prolonged farrowing duration (≥300 min) (Figure 1). Pearson’s correlations between farrowing duration, birth interval and sow reproductive characteristics are shown in Table 2. Farrowing duration was positively correlated with TB (r = 0.141, *p* < 0.001), SB (r = 0.259, *p* < 0.001), MF (r = 0.049, *p* = 0.015), piglet birth weight (r = 0.068, *p* < 0.001) and litter birth weight (r = 0.041, *p* = 0.043).

### 3.2. Effect of Litter Size

The reproductive performance and farrowing characteristics of sows that had 8–12, 13–15 and ≥16 TB are presented in Table 3. The proportion of sows with a prolonged farrowing in the litters that had TB ≥ 16 was higher than that in litters with TB 8–12 (*p* < 0.001) and tended to be higher than that in litters with TB 13–15 (*p* = 0.071). Likewise, the farrowing duration of sows that had TB ≥ 16 and 13–15 was longer than that of sows with TB 8–12 (Table 3). However, the birth interval of sows that had TB ≥ 16 was shorter than that of sows that had TB 13–15 and 8–12 (Table 3). In addition, the SB was higher and the piglet birth weight lower in the litters with ≥16 TB compared to the litters with 8–12 TB (Table 3). Farrowing duration and birth intervals among the TB groups by parity classes are presented in Figure 2a,b, respectively. Prolonged farrowing as well as a long birth interval were frequently detected in sow parities 5–7 and 8–10. In addition, farrowing duration and birth interval in sows that had 3 and ≥4 stillborn piglets per litter were longer than in sows that had 0, 1 and 2 stillborn piglets per litter (Figure 3).

### 3.3. Effect of Parity Numbers

The farrowing duration differed among the parity groups (*p* < 0.05). The farrowing duration of sows with parity numbers 5–7 and 8–10 was longer than that of sows with parity numbers 1 and 2–4 (Table 4). Primiparous sows that had TB 8–12 had the shortest farrowing duration (173.2 ± 8.8 min), and sows with parity numbers 5–7 and TB ≥ 16 had the longest farrowing duration (263.2 ± 10.0 min) (Figure 2a). Birth intervals also differed among parities. Birth intervals of sows with parity numbers 5–7 and 8–10 were longer than those of sows with parity numbers 1 and 2–4 (Table 4).

### 3.4. Onset of Farrowing

Overall, farrowing duration, birth intervals and SB were affected by the onset of farrowing (*p* < 0.05). In general, sows that started farrowing during working hours had a longer farrowing duration (229.3 ± 3.6 min) than sows that started farrowing during non-working hours (217.6 ± 3.4 min, *p* = 0.017). Similarly, sows that started farrowing during working hours also had longer birth intervals (17.0 ± 0.3 min) than sows that started farrowing during non-working hours (16.1 ± 0.3 min, *p* = 0.019). On the other hand, sows that started farrowing during working hours had a lower SB than sows that started to farrow during non-working hours (5.8 ± 0.2% vs. 6.7 ± 0.2%, *p* = 0.007). The farrowing duration, birth intervals and SB during working hours and non-working hours in different classes of TB and parity number are presented in Table 5. Interestingly, the difference in the farrowing duration of sows between sows that started farrowing during working hours and non-working hours was significant in only sow parity numbers 5–7 (Table 5). Similarly, the influence of onset of farrowing on the birth interval of piglets was also detected in sow parity numbers 5–7 (Table 5).

### 3.5. Temperature and Humidity

In the present study, the housing had a good cooling system, resulting in a narrow temperature range (24.2–27.3 °C) and stable relative humidity (69.7%–72.1%) inside the barn. In general, only the average maximum temperature and the maximum temperature–humidity index in the farrowing house for a period of 7 days before farrowing influenced farrowing duration (*p* < 0.05). Farrowing duration in primiparous sows was not correlated with daily mean temperature, daily maximum temperature, relative humidity, temperature–humidity index, or maximum temperature–humidity index (*p* > 0.05) (Table 6). However, in multiparous sows, farrowing duration was positively correlated with maximum temperature (r = 0.056, *p* = 0.012) and maximum temperature–humidity index (r = 0.059, *p* = 0.008) during the 7-day period before farrowing (Table 6).

## 4. Discussion

### 4.1. Effect of Litter Size

In this study, the sows in a commercial swine herd in Thailand currently had 13.7 TB. This indicates that the TB has increased by 38% over the last two decades (TB 9.9) [19]. In another commercial swine herd in Thailand, the average TB was as high as 17.5 [4]. This is mainly due to the import of modern hyperprolific sows from European countries, especially from Denmark. When classifying the litters according to TB, sows that had ≥16 TB under the tropical climate accounted for more than a quarter of the sow population. This indicates that sows with ≥16 TB are becoming increasingly common in the swine industry. We demonstrated that farrowing duration increased following an increase in TB, while the average birth interval decreased. This can be due to the routine administration of oxytocin in all sows after the birth of the 10th piglet. The administration of oxytocin during parturition can increase the duration and intensity of myometrium contraction, thus decreasing farrowing duration and the average birth interval [20]. However, the proportion of sows with a prolonged farrowing duration (i.e., >300 min) with litters of TB ≥ 16 was higher than that with a lower TB. Parturition may last longer in large litters because of the accumulation of expulsion for each piglet [21]. In sows as well as in other species, labour is suspected to be painful due to contraction of the uterus, foetal expulsion, and female reproductive tract inflammation, especially if it lasts for more than 3 h [9]. Therefore, an endogenous opioid-mediated analgesia system exists as a defence mechanism against pain during parturition [21]. However, increasing the release of opioids due to severe pain and stress can interfere with oxytocin [9,21], especially during farrowing in sows with large litters. This is supported by a study in rats, which found an opioid-dependent reduction of oxytocin release during prolonged parturition under stress [22]. There are two known mechanisms for the inhibition of oxytocin release by opioids. First, opioids bind to κ-opioid receptors in the neurohypophysis, which results in the inhibition of neurosecretory terminals [23]; second, opioids bind to μ-opioid receptors in the paraventricular nucleus, resulting in a reduction in the pulse rate of oxytocinergic neurons [24]. Moreover, pain during parturition also activates the autonomic nervous system, which increases catecholamine secretion. High plasma concentration of catecholamine has been considered to affect uterine motility by reducing myometrial contractibility and promoting muscular relaxation. These mechanisms can lead to prolonged farrowing and increased number of nociceptive signals [25]. Decreasing the oxytocin secretion can reduce uterine contraction and results in a prolonged piglet expulsion. In addition, parturition requires energy, and in large litters, the energy demand may be greater. Uterine and mother fatigue due to insufficient energy can cause delivery difficulties or even stop farrowing in sows [21]. Thus, sows with large litters are more susceptible to experiencing severe pain and stress, leading to a decrease in oxytocin release. Moreover, insufficient maternal energy in sows with large litters during parturition may lead to slowing down uterine contractions, further prolonging the farrowing duration. In the present study, sows with prolonged farrowing had more stillborn piglets than sows with shorter farrowing duration. Therefore, various procedures to increase the uterine contractions of sows with large litters need to be comprehensively investigated [26].

### 4.2. Effect of Parity Number

In the present study, farrowing duration and birth intervals were longer in old sows (parity numbers 5–7 and 8–10) than in young sows (parity numbers 1 and 2–4). This agrees with a previous study that demonstrated longer farrowing durations in sows with higher parity numbers compared to those with lower ones [27]. On the other hand, van Dijk et al. [28] demonstrated that parity number did not affect the duration of the expulsive stage. Additionally, Yang et al. [29] found a shorter farrowing duration in sows with higher parity numbers (6–9) compared to sows with lower parity numbers (i.e., 1 and 2–5). However, in their study, sows with parity numbers 6–9 had a smaller total number of piglets born per litter (12.4 ± 3.2) than sows with parity numbers 2–5 and 1 (15.5 ± 3.1 and 16.8 ± 1.8) [29]. Litter size is positively correlated with farrowing duration, and thus, in sows with large litters, farrowing may last longer. In the present study, farrowing duration and birth intervals were longer in sows with higher parity numbers within the same classes of the total number of piglets born per litter. It is suspected that primiparous sows are more susceptible to experiencing painful parturition than multiparous sows [9], with effects on the duration of labour. However, a previous study demonstrated that sows experienced more pain than gilts due to the uterine activity during farrowing [30]. Ison et al. [30] found that sows showed more frequent back arching as an indicator of pain and had higher salivary cortisol levels than gilts on the day of farrowing. In addition, a study in rats demonstrated that young rats exhibited greater spontaneous contractile activity in myometrium tissue and tended to be stronger than during labour than old rats [31]. Also, integral activity and rate of contraction were greater in young rats than in old rats [31]. Similarly, Mota-Rojas et al. [25] reported that gilts had a better uterine contraction and greater contraction intensity than old sows (i.e., sixth parity). This supports our finding that farrowing duration in sows with lower parity numbers is shorter than in those with higher parity numbers. With less pain during parturition and primed myometrium, the duration of labouring is expected to be shorter in younger animals. These findings indicate that prolonged farrowing duration in hyperprolific sows is a serious issue in multiparous rather than primiparous sows. Pain management as well as improved myometrial activity in multiparous sows should therefore be focused on.

### 4.3. Onset of Farrowing

In the modern swine industry, farrowing management is important to optimise pig production, with sows starting farrowing during working hours becoming preferable compared to those starting farrowing during non-working hours. In the present study, we found that the farrowing duration and birth interval lasted longer when the sows had started farrowing during working hours compared to sows that had started farrowing during non-working hours. Surprisingly, even though the farrowing duration was longer, the stillbirth rate was lower when sows had started farrowing during working hours. This can be explained by the noise in the farrowing house during working hours, either from the sows themselves (e.g., screaming during feeding times) or the environment (e.g., mechanical ventilation, high-pressure cleaning, feed mixing and manure removal lines) [32]. A previous study found that pigs are sensitive to prolonged or intermittent noise, which can cause increased cortisol levels [33]. A previous study in women found that cortisol and oxytocin levels are reciprocal: when cortisol increases, then oxytocin decreases and vice versa [34]. Therefore, sows that start farrowing during working hours are likely susceptible to noise stress, which can lead to an increase in cortisol levels and a decrease in oxytocin levels, resulting in delayed foetal expulsion. A previous study also confirmed that maternal stress can lead to hyperactivity of the hypothalamic–pituitary–adrenal axis and the sympathetic–adrenal–medullary system, which is related to catecholamine release [25]. As stated before, catecholamine compromises uterine activity during parturition. However, farrowing during working hours is preferable due to the ease of monitoring, and staff can assist immediately if dystocia occurs. In a previous study, the administration of exogenous hormones to control the onset of farrowing was investigated [35,36]. The authors demonstrated that altrenogest supplementation in combination with double administrations of PGF2ɑ successfully synchronised the onset of farrowing in sows, with the proportion of sows that started farrowing during working hours tending to be higher in the treatment group than in the control group. However, the use of pharmacology to control parturition in sows needs to be carefully considered due to its side effects, e.g., reduced colostrum intake and high stillbirth rates [35].

### 4.4. Temperature and Humidity

Interestingly, the maximum temperature and maximum temperature–humidity index were positively correlated with farrowing duration in multiparous sows (parity number 2–10), but this was not the case in primiparous sows. In contrast, Iida et al. [18] demonstrated that gilts had a lower critical temperature threshold than sows, making them more susceptible to heat stress than sows. In a previous study, sows kept at lower room temperatures (15 °C) had shorter farrowing durations and birth intervals than sows kept at higher room temperatures (20 and 25 °C) [37]. Similarly, Muns et al. [38] demonstrated that second-parity sows kept at an initial room temperature of 20 °C, with a gradual increase to 25 °C, from d 112 to 115 of gestation had a longer farrowing duration compared to sows kept at a room temperature of 20 °C. Failure to perform thermoregulatory behaviour around farrowing may result in heat stress, which has negative effects that lead to prolonged farrowing [38]. Pigs require different housing temperatures for different reproductive stages. Gestating sows need housing temperatures of 15–24 °C, whereas lactating sows require slightly lower housing temperatures: 15–21 °C [39]. However, suckling piglets need much higher housing temperatures: 28–32 °C [39]. Thus, in addition to the mother’s requirements, it is important to provide a heater behind the sows during the first few hours after birth, which can reduce hypothermia in newborn piglets. Our findings indicate that heat stress due to rising temperatures and/or humidity in the 7-day period before parturition can cause prolonged farrowing durations in multiparous sows. However, temperature or humidity may not be the only factors influencing the farrowing duration of sows, since they have a relatively low correlation coefficient. Therefore, other factors, such as parity number and maternal stress, may also contribute to the prolonged farrowing duration problem in sows under tropical conditions.

## 5. Conclusions

The significant risk factors associated with the farrowing duration of Landrace x Yorkshire sows kept in free farrowing pens under tropical conditions included TB, SB, MF, parity number, litter birth weight, piglet birth weight, and time of onset of farrowing. The proportion of sows with a prolonged farrowing in the litters that had TB ≥ 16 was higher than that in litters with TB 8–12. Sows that started farrowing during working hours had a longer farrowing duration than sows that started farrowing during non-working hours. The farrowing duration of sows with parity numbers 5–7 and 8–10 was longer than that of sows with parity numbers 1 and 2–4. For multiparous sows, the maximum temperature and the maximum temperature–humidity index also influenced the farrowing duration.

## Figures and Tables

**Figure 1 animals-12-02943-f001:**
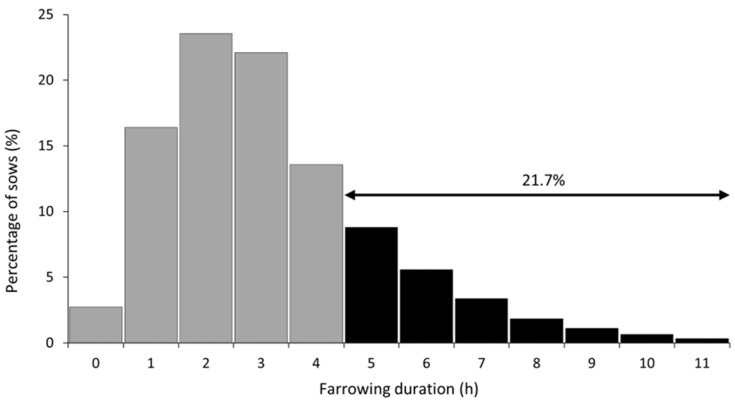
Frequency distribution of sows based on farrowing duration (*n* = 2493).

**Figure 2 animals-12-02943-f002:**
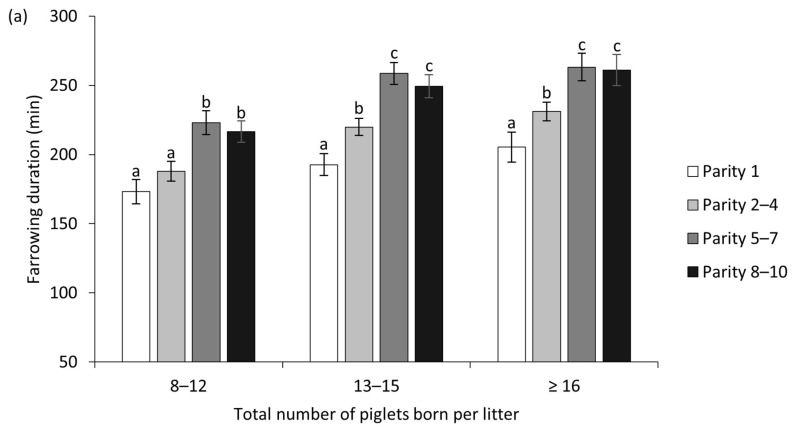
The duration of farrowing (**a**) and birth intervals (**b**) of sows among the total number of piglets born per litters and parities classes. ^a,b,c^ Different lowercase letters within the class of total number of piglets born per litter denote data that differ significantly (*p* < 0.05). Bars indicate standard error.

**Figure 3 animals-12-02943-f003:**
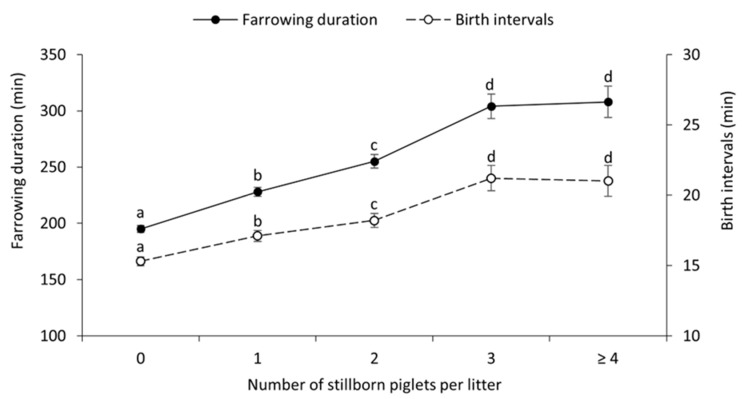
Farrowing duration and birth intervals of sows with different numbers of stillborn piglets per litter. ^a,b,c,d^ Different superscripts differ significantly (*p* < 0.05).

**Table 1 animals-12-02943-t001:** Descriptive statistics on reproductive performance and farrowing characteristics of 2493 Landrace x Yorkshire cross-bred sows in a commercial swine herd in Thailand.

Variables	Means ± SD	Range
Parity number	4.4 ± 2.9	1–10
Total number of piglets born per litter	13.7 ± 2.8	8–23
Number of piglets born alive per litter	12.1 ± 3.1	0–23
Stillborn piglets per litter (%)	5.9 ± 8.0	0–100
Mummified foetuses per litter (%)	5.8 ± 12.4	0–100
Litter birth weight (kg)	16.2 ± 4.0	1.9–41.4
Piglet birth weight (kg)	1.4 ± 0.2	0.8–3.2
Piglet with birth weight <1.0 kg (%)	11.0 ± 13.4	0–87.5
Duration of farrowing (min)	221.0 ± 119.3	30.0–716.0
Birth interval (min)	16.6 ± 9.4	2.1–71.9

**Table 2 animals-12-02943-t002:** Pearson’s correlations between farrowing duration (mean ± SD = 221.0 ± 119.3 min), birth interval (mean ± SD = 16.6 ± 9.4 min) and sow reproductive characteristics (*n* = 2493).

Variables	Correlation Coefficient (r)	
	Farrowing Duration	Birth Interval
Total number of piglets born per litter	0.141 ***	−0.239 ***
Stillborn piglets per litter (%)	0.259 ***	0.167 ***
Mummified foetuses per litter (%)	0.049 **	NS
Piglet birth weight (kg)	0.068 ***	0.181 ***
Litter birth weight (kg)	0.041 *	−0.177 ***
Piglets with birth weight <1.0 kg	NS	−0.065 ***
Number of piglets born alive per litter	NS	−0.280 ***

NS = not significant (*p* > 0.05); significance is indicated by * 0.01 < *p* < 0.05, ** 0.001 < *p* < 0.01 and *** *p* < 0.001.

**Table 3 animals-12-02943-t003:** Reproductive performances and farrowing characteristics of sows by the total number of piglets born per litter.

Variables	Total Number of Piglets Born per Litter		
	8–12	13–15	≥16
Number of sows	853	983	657
Parity number	4.7 ± 0.1 ^a^	4.3 ± 0.1 ^b^	4.0 ± 0.1 ^c^
Total number of piglets born per litter	10.6 ± 0.0 ^a^	14.0 ± 0.0 ^b^	17.1 ± 0.0 ^c^
Number of piglets born alive per litter	9.4 ± 0.1 ^a^	12.4 ± 0.1 ^b^	15.1 ± 0.1 ^c^
Stillborn piglets per litter (%)	5.5 ± 0.3 ^a^	6.0 ± 0.3 ^a,b^	6.4 ± 0.3 ^b^
Mummified foetuses per litter (%)	6.0 ± 0.4 ^a^	5.9 ± 0.4 ^a^	5.4 ± 0.5 ^a^
Litter birth weight (kg)	13.6 ± 0.1 ^a^	16.6 ± 0.1 ^b^	19.2 ± 0.1 ^c^
Piglet birth weight (kg)	1.43 ± 0.01 ^a^	1.34 ± 0.01 ^b^	1.27 ± 0.01 ^c^
Piglets with birth weight <1.0 kg (%)	7.8 ± 0.5 ^a^	10.9 ± 0.4 ^b^	15.4 ± 0.5 ^c^
Farrowing duration (min)	199.9 ± 4.0 ^a^	228.2 ± 3.8 ^b^	237.9 ± 4.6 ^b^
Proportion of sows farrowed >300 min (%)	17.0 ^a^	22.9 ^b^	25.9 ^b^
Birth interval (min)	19.1 ± 0.3 ^a^	16.4 ± 0.3 ^b^	13.9 ± 0.4 ^c^

^a,b,c^ Different superscript letters within a row denote data that differ significantly (*p* < 0.05).

**Table 4 animals-12-02943-t004:** Reproductive performance and farrowing characteristics of sows by parity number.

Variables	Parity Number			
	1	2–4	5–7	8–10
Number of sows	506	919	534	534
Total number of piglets born per litter	13.4 ± 0.1 ^a^	14.1 ± 0.1 ^b^	13.7 ± 0.1 ^a^	13.0 ± 0.1 ^c^
Number of piglets born alive per litter	12.3 ± 0.1 ^a^	12.8 ± 0.1 ^b^	11.7 ± 0.1 ^c^	10.9 ± 0.1 ^d^
Stillborn piglets per litter (%)	3.8 ± 0.3 ^a^	4.4 ± 0.3 ^a^	7.6 ± 0.3 ^b^	8.9 ± 0.3 ^c^
Mummified foetuses per litter (%)	4.5 ± 0.5 ^a^	4.9 ± 0.4 ^a^	7.2 ± 0.5 ^b^	7.2 ± 0.5 ^b^
Litter birth weight (kg)	15.6 ± 0.2 ^a^	17.4 ± 0.1 ^b^	16.0 ± 0.2 ^a^	15.1 ± 0.2 ^c^
Piglet birth weight (kg)	1.28 ± 0.01 ^a^	1.37 ± 0.01 ^b^	1.36 ± 0.01 ^b^	1.38 ± 0.01 ^b^
Piglets with birth weight <1.0 kg (%)	12.1 ± 0.6 ^a^	10.0 ± 0.4 ^b^	12.0 ± 0.6 ^a^	10.9 ± 0.6 ^a,b^
Farrowing duration (min)	188.3 ± 5.2 ^a^	214.3 ± 3.9 ^b^	247.7 ± 5.1 ^c^	237.1 ± 5.1 ^c^
Proportion of sows farrowed >300 min (%)	13.8 ^a^	18.3 ^a^	29.6 ^b^	27.0 ^b^
Birth interval (min)	14.5 ± 0.4 ^a^	15.5 ± 0.3 ^b^	18.6 ± 0.4 ^c^	18.7 ± 0.4 ^c^

^a,b,c,d^ Different superscript letters within a row denote data that differ significantly (*p* < 0.05).

**Table 5 animals-12-02943-t005:** Farrowing duration, birth intervals and percentage of stillborn piglets during working hours (07.00 h–17.00 h) and non-working hours (17.01 h–06.59 h) by total number of piglets born per litter (TB) and parity number.

Variables	Working Hours	Non-Working Hours	*p* Value
Number of sows	1176	1317	
Farrowing duration (min)			
All sows	229.3 ± 3.6	217.6 ± 3.4	0.017
TB classes			
8–12	206.2 ± 5.9	194.1 ± 5.5	NS
13–15	236.3 ± 5.6	223.9 ± 5.2	NS
≥16	245.5 ± 6.9	234.9 ± 6.7	NS
Parity number classes			
1	194.9 ± 7.8	185.9 ± 7.2	NS
2–4	215.5 ± 5.5	210.4 ± 5.4	NS
5–7	260.7 ± 7.1	235.8 ± 7.3	0.014
8–10	246.2 ± 7.9	238.4 ± 6.9	NS
Birth interval (min)			
All sows	17.0 ± 0.3	16.1 ± 0.3	0.019
TB classes			
8–12	19.6 ± 0.5	18.6 ± 0.4	NS
13–15	17.0 ± 0.4	16.0 ± 0.4	NS
≥16	14.4 ± 0.5	13.8 ± 0.5	NS
Parity number classes			
1	14.7 ± 0.6	13.9 ± 0.6	NS
2–4	15.7 ± 0.4	15.4 ± 0.4	NS
5–7	19.3 ± 0.6	17.3 ± 0.6	0.010
8–10	18.4 ± 0.6	17.9 ± 0.5	NS
Stillborn piglets (%)			
All sows	5.8 ± 0.2	6.7 ± 0.2	0.007
TB classes			
8–12	4.9 ± 0.4	6.1 ± 0.4	0.025
13–15	5.6 ± 0.4	6.9 ± 0.3	0.012
≥16	6.8 ± 0.5	7.0 ± 0.4	NS
Parity number classes			
1	3.3 ± 0.5	4.4 ± 0.5	NS
2–4	4.2 ± 0.4	4.5 ± 0.4	NS
5–7	7.6 ± 0.5	7.6 ± 0.5	NS
8–10	7.9 ± 0.5	10.1 ± 0.5	0.001

NS = not significant (*p* > 0.05).

**Table 6 animals-12-02943-t006:** Pearson’s correlations between farrowing duration and climatic parameters during the 7-day period before farrowing in sows by parity number.

Climatic Parameters	Mean ± SD	Parity Number of Sows	
		Primiparous	Multiparous (Parities 2–10)
Number of sows		506	1987
Mean temperature (°C)	26.5 ± 0.3	NS	NS
Maximum temperature (°C)	28.4 ± 0.3	NS	0.056 *
Relative humidity (%)	71.0 ± 0.2	NS	NS
Temperature-humidity index	76.4 ± 0.4	NS	NS
Maximum temperature-humidity index	79.2 ± 0.4	NS	0.059 **

NS = not significant (*p* > 0.05); significance levels are indicated by * 0.01 < *p* < 0.05, ** 0.001 < *p* < 0.01.

## Data Availability

The data presented in this study are available on request from the corresponding author. The data are not publicly available due to the original data belonging to the swine breeding herd.

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
