# Peer review of "Factors Associated with Farrowing Duration in Hyperprolific Sows in a Free Farrowing System under Tropical Conditions"

_animals, 2022, doi:10.3390/ani12212943_

Round 1
Reviewer 1 Report
This manuscript talks about an important topic in swine production.
The introduction has important aspects but, I suggest summarizing the information a make it concise
Materials and methods
Numerals 2.1 and 2.3 can be merged
Results
I suggest removing Fig 1, because it is giving similar information than is in table 5
Line 126 -127> the idea is not clear when say ”recorded and merged with…”
Line 217 – 220. Data are presented in the table, maybe it is not necessary to write in this way.
Line 250 and line 270: There are Fig 3. a and b, please describe the text
Numeral 3.3 “Effects of parity numbers” I respectfully consider a different sub-title i.e. parameters…
The description of results on numeral 3.4, refers to table 5 but the description is not completely related to the table, I suggest checking the numbers and significance analysis
Discussion
Line 303, I suggest referring to this particular study.
Line 320 – 329: I suggest removing this section; because it is not related to the results or variables evaluated in this study.
Line 397-401: I suggest to add these lines to results section, and focusing on discussing the main topic.
The conclusion can be improved, highlighting the factors that were significantly related to farrowing duration.
Author Response
Comments and Suggestions for Authors
This manuscript talks about an important topic in swine production. The introduction has important aspects but, I suggest summarizing the information a make it concise.
ANSWER: Thank you very much for your kindly consideration. The manuscript has been revised according to the suggestion made by all reviewers. Any changes in the manuscript is indicated by using ‘blue text’.
Materials and methods
Numerals 2.1 and 2.3 can be merged
Answer: Subtitle 2.3 has been changes to 2.2 following ‘2.1 Study designs’. The name has been changed to be ‘2.2 Data: inclusion and exclusion criteria’ according to suggestion made by reviewer 2. Thank you very much.
Results
I suggest removing Fig 1, because it is giving similar information than is in Table 5.
ANSWER: Figure 1 has been deleted. An additional sentence has been added in text. “The proportion of sows with the onset of farrowing during working hours and non-working hours were 47.2% and 52.8%, respectively.”
Line 126 -127: the idea is not clear when say “…recorded and merged with…”
ANSWER: We deleted the word “and merged with the farrowing data” to make it simple. The idea is that “the average temperature, humidity and temperature-humidity index (THI) during the 7-day period before farrowing was calculated and then added to the farrowing records of each individual sows.”
Line 217 – 220. Data are presented in the table, maybe it is not necessary to write in this way.
ANSWER: The sentence has been modified. Some descriptive data from text has been deleted and refer to Table 1 instead. Thank you.
Line 250 and line 270: There are Fig 3. a and b, please describe the text
ANSWER: Fig 3 a and b have been added into the text.
Numeral 3.3 “Effects of parity numbers” I respectfully consider a different sub-title i.e. parameters…
ANSWER: We are willing to demonstrate the influence of parity number in this section. Therefore, the subtitle ‘Effect of parity numbers’ was kept. Thank you for your advice.
The description of results on numeral 3.4, refers to Table 5 but the description is not completely related to the Table, I suggest checking the numbers and significance analysis.
ANSWER: We rewrote some part to make it clearer. The description of results on numeral 3.4 explain about the difference of farrowing duration, birth intervals and SB in general between the sows that start to farrow during the working hours and non-working hours. However, in Table 5 we explained the same aspect but within the classes of total born (TB) and parity number. However, additional information from Table 5 has been added.
Discussion
Line 303, I suggest referring to this particular study.
ANSWER: We rewrote the sentence as suggested.
Line 320 – 329: I suggest removing this section; because it is not related to the results or variables evaluated in this study.
ANSWER: We have added some additional information to make it clearer according to comments made by Reviewer #2. Thank you.
Line 397-401: I suggest to add these lines to results section, and focusing on discussing the main topic.
ANSWER: We have moved those lines to the result section as suggested. Thank you.
The conclusion can be improved, highlighting the factors that were significantly related to farrowing duration.
ANSWER: We have added some highlights of the factors that were significantly related to farrowing duration as suggested: ……“The proportion of sows with a prolonged farrowing in the litters that had TB ≥ 16 was higher than that in litters with TB 8–12. Sows that started farrowing during working hours had a longer farrowing duration than sows that started farrowing during non-working hours. The farrowing duration of sows with parity numbers 5–7 and 8–10 was longer than that of sows with parity numbers 1 and 2–4.”
Reviewer 2 Report
Pregnancy and farrowing are critical periods in which both the mother and the fetus are exposed to physiological responses that alter their homeostasis. Particularly in sows, one of the problems that have been studied is the prolonged duration of farrowing, since this aspect can cause postpartum complications and increase the percentage of stillborn piglets per litter. The present study provides valuable information by investigating the factors that influence the farrowing duration and the proportion of hyperprolific sows who had prolonged labor maintained in a free delivery system in a tropical environment, resulting in a significant advance in the area of perinatology and reproduction. However, some points must be addressed to achieve publication quality. I have left some comments hoping that they can help the authors, but not before congratulating them on the orderly presentation of the manuscript.
General Comments
Line 70: the idea “Currently, the average farrowing duration in hyperprolific sows in Europe is longer than 300 min” is repetitive with lines 56-61, please delete.
Lines 132-133: In the study design, it is only necessary to list or mention the factors associated with the duration of farrowing that were included as variables. The information presented refers to statistical analysis, so these lines must be rewritten or placed in the corresponding section.
Lines 141-144: please indicate the percentage of crude protein and metabolizable energy of the food used.
Line 149: the idea “The farrowing process was carefully monitored” is ambiguous, a more detailed explanation of how the farrowing was monitored is required, so that the reader has a better understanding of what was done.
Lines 151-153: Please indicate the route of administration of the oxytocin used in sows without signs of uterine contraction and in those animals in which it was used routinely. Please clarify if oxytocin was used in a single dose in the first case.
Line 171: section 2.3; I suggest that it be named as follows,
Data: inclusion and exclusion criteria
Please also clarify if sows without signs of uterine contraction or those that received a higher dose of oxytocin were also excluded from the study?
Line 188: please remove the dot inside the parentheses.
Table 2 and 6: Although there is a significant difference in the study variables, the degree of correlation calculated is low. Therefore, I suggest a deeper discussion of these results, to explain the low correlation.
Discussion. In the effect of litter size section (4.1), it is not only endogenous opioids that affect oxytocin neurosecretion. There are other mechanisms involved that should be discussed, for example, the noradrenergic and catecholamine secretion pathways, cortisol, and GABAergic neurons, including also in section 4.3 on the onset of farrowing. I suggest reviewing this document to supplement your discussion: https://doi.org/10.3390/ani12192686
Author Response
Reviewer #2
Pregnancy and farrowing are critical periods in which both the mother and the fetus are exposed to physiological responses that alter their homeostasis. Particularly in sows, one of the problems that have been studied is the prolonged duration of farrowing, since this aspect can cause postpartum complications and increase the percentage of stillborn piglets per litter. The present study provides valuable information by investigating the factors that influence the farrowing duration and the proportion of hyperprolific sows who had prolonged labor maintained in a free delivery system in a tropical environment, resulting in a significant advance in the area of perinatology and reproduction. However, some points must be addressed to achieve publication quality. I have left some comments hoping that they can help the authors, but not before congratulating them on the orderly presentation of the manuscript.
ANSWER: Thank you very much for your kindly consideration. The comments have been followed carefully. The manuscript has been revised point-by-point according to the reviewer’s comments. Responses to each specific comment are listed below.
General Comments
Line 70: The idea “Currently, the average farrowing duration in hyperprolific sows in Europe is longer than 300 min” is repetitive with lines 56-61, please delete.
ANSWER: Agree. The sentence has been deleted as suggested.
Lines 132-133: In the study design, it is only necessary to list or mention the factors associated with the duration of farrowing that were included as variables. The information presented refers to statistical analysis, so these lines must be rewritten or placed in the corresponding section.
ANSWER: The information presented refers to statistical analysis has been deleted. Additional information concerning variables included in the analyses has been added.
Lines 141-144: please indicate the percentage of crude protein and metabolizable energy of the food used.
ANSWER: The gestation diet contained 15.0% crude protein, 2,700 kcal/kg metabolizable energy, and 0.7% lysine. The lactation diet contained 16.0% crude protein, 3,600 kcal/kg metabolizable energy, and 0.8% lysine.
Line 149: the idea “The farrowing process was carefully monitored” is ambiguous, a more detailed explanation of how the farrowing was monitored is required, so that the reader has a better understanding of what was done.
ANSWER: We have added additional details to explain how monitoring was done.
Lines 151-153: Please indicate the route of administration of the oxytocin used in sows without signs of uterine contraction and in those animals in which it was used routinely. Please clarify if oxytocin was used in a single dose in the first case.
ANSWER: We added the route of administration (IM) of the oxytocin in the text. We clarified that oxytocin used in the first case was a single dose administration.
Line 171: section 2.3; I suggest that it be named as follows, Data: inclusion and exclusion criteria
ANSWER: We change the name of the section as suggested. This section has been moved to be section ‘2.2. Data: inclusion and exclusion criteria’. Thank you.
Please also clarify if sows without signs of uterine contraction or those that received a higher dose of oxytocin were also excluded from the study?
ANSWER: We did not excluded sows that received oxytocin from the study because all of the sows received a routine administration of oxytocin after the 10th piglets was born. Moreover, the sows received extra doses of oxytocin did not precisely recorded.
Line 188: please remove the dot inside the parentheses.
ANSWER: The dot has been removed. Thank you.
Table 2 and 6: Although there is a significant difference in the study variables, the degree of correlation calculated is low. Therefore, I suggest a deeper discussion of these results, to explain the low correlation.
ANSWER: The low correlation coefficient could be due a high number of observations included in the analyses and also a high variation among sows due to some other factors and random error. However, additional discussion has also been addressed i.e., “However, either temperature or humidity may not be the only single factor influencing the farrowing duration of sows since they have a relatively low correlation coefficient. Therefore, some other factors, such as parity number and maternal stress, may also contribute to the prolonged farrowing duration problem in sows under tropical conditions.”
Discussion
In the effect of litter size section (4.1), it is not only endogenous opioids that affect oxytocin neurosecretion. There are other mechanisms involved that should be discussed, for example, the noradrenergic and catecholamine secretion pathways, cortisol, and GABAergic neurons, including also in section 4.3 on the onset of farrowing. I suggest reviewing this document to supplement your discussion: https://doi.org/10.3390/ani12192686
ANSWER: The article has been reviewed and additional discussion concerning these issues has been added. This reference has also been added in the reference list. Thank you very much.
Reviewer 3 Report
Well-written and no major edits, the only point I suggest checking throughout is for implications of causation of factors on farrowing duration from the results of the study. Since this was a retrospective study, correlation/relationships between factors and farrowing duration can only be described as related rather than causative, although it can be suggested in discussion using other references that found causative relationships as support.
Author Response
Reviewer #3
Well-written and no major edits, the only point I suggest checking throughout is for implications of causation of factors on farrowing duration from the results of the study. Since this was a retrospective study, correlation/relationships between factors and farrowing duration can only be described as related rather than causative, although it can be suggested in discussion using other references that found causative relationships as support.
Answer: Thank you very much for your kindly consideration. The manuscript has been revised according to the suggestion.